

# Usability and motivational impact of a fast-paced immersive virtual reality lecture on international middle school students in geoscience education

Azim ZULHILMI[1], Yuichi S. HAYAKAWA[2], Daniel R. NEWMAN[2]

[1]Graduate School of Environmental Science, Hokkaido University, Sapporo, Japan

[2]Faculty of Environmental Earth Science, Hokkaido University, Sapporo, Japan

*Correspondence to*: Azim Zulhilmi (Azimzulhilmi@ees.hokudai.ac.jp)

**Abstract.** Immersive Virtual Reality (VR) offers educators an innovative tool to teach geoscience, addressing challenges in conveying the Earth's 3D characteristics traditionally
taught through field visits and experiences that are often inaccessible to many due to socioeconomic, political, and logistical barriers. VR provides an alternative experience, allowing users to virtually explore geological sites beyond physical and situational constraints. Despite its potential, the implementation of geoscience-focused VR lectures remains largely unexplored. As a pioneering case study, this research investigates the usability and motivational
impact of VR by developing a fast-paced virtual reality lecture on landslides for middle school students. Approximately 60 students from diverse cultural and educational backgrounds participated. Results revealed that the VR lecture was usable, with key strengths in its ability to engage students and deliver satisfaction. Compared to traditional teaching methods (lecture and hands-on), VR excelled in fostering interest, enjoyment, and perceived choice. This study
provides valuable insights into the practical implementation of VR in geoscience education, demonstrating its potential to make geoscience topics more accessible and engaging for diverse student groups. Future research should explore strategies to address usability challenges and enhance the motivational attributes of VR, paving the way for its broader adoption in geoscience educational settings.

## 1 Introduction

Geoscience education can be referred as the structured teaching and learning that focuses on the earth's physical features, processes, and systems. It is important that when teaching geoscience, students may need to develop observational and spatial thinking skills, as it is an observational science. In particular, field experiences in geoscience require a level of thinking,
mental visualization, and investigative skills not commonly found in other scientific fields (King, 2008; Liben et al. 2011). Therefore, it is necessary for educators of geoscience to emphasize on transferring these key skills to their students.

Conventional geoscience teaching methods using textbooks and lectures often fail to effectively convey the reality of the nature, including three-dimensional (3D) features of the
Earth. These approaches tend to rely on pseudo-3D visualizations to express the earth which is challenging because it places additional cognitive burden on learners to mentally visualize complex 3D geometrical concepts that they may not be very familiar with (Fitzpatrick & Hedley, 2024; Havenith et al. 2019). Hands-on activities like fieldwork, museum visits, and lab activities help address this limitation, but access is often restricted due to practical issues like
finance, health, socioeconomic, and logistical barriers. Addressing these challenges is vital for advancing geoscience education for the masses.



Immersive Virtual Reality (VR) offers a promising solution to tackle this issue. VR provides an immersive, interactive, and realistic 3D experience accessible through head-mounted displays (HMD) and their respective peripherals. It addresses the accessibility and visualization challenges of traditional geoscience teaching methods by enabling fully immersive realistic exploration of geological data in 3D, with intuitive motion-based interaction while also leveraging physical and cyber portability. This makes VR a potentially versatile and highly useful tool for geoscience education and communication.

### 1.1 Literature Review

One of the major global challenges today is ensuring quality education. As outlined by the Sustainable Development Goal 4 in United Nations (2015):

> "Ensure inclusive and equitable quality education and promote lifelong learning opportunities for all" (p. 14).

The geoscience community has made significant strides in communicating geoscience to the broader public, facilitating knowledge dissemination beyond the scientific community. While traditional communication methods, such as lectures, museums, and workshops, have been widely utilized, advancements in technology now provide more innovative outreach approaches.

Studies on 3D printing in geological education highlight its benefits: Gutierrez et al. (2023) demonstrated that 3D-printed models improve undergraduates' spatial and visualization skills while enhancing understanding of geological structures. Similarly, Chenrai (2021) found that integrating 3D printing into geoscience curricula bolstered structural interpretation and spatial visualization abilities.

Virtual Field Trips (VFTs) are effective tools for geological outreach: Watson et al. (2022) found personal computer (PC)-based VFTs engaging for teaching physical volcanology through 3D visuals and videos. Tibaldi et al. (2020) described VR approach using 3D digital outcrop models for teaching, learning, and research in volcanology suggest that VR is capable for engaging public and can be value to promote environmental site protection and development. De Paz-Álvarez et al. (2022) demonstrated VFTs' usefulness in teaching mapping skills but noted they cannot fully replace traditional courses. Klippel et al. (2019) showed immersive VFTs with VR headsets improved lab performance and enthusiasm compared to traditional field trips.

Other VR applications show promise in geoscience communication: Yamauchi et al. (2022) used VR to visualize underground heritage (Taya Cavern in Yokohama, Japan), boosting interest and encouraging potential real-world visits. Alene et al. (2024) developed "QuickAware," a VR tool raising awareness of quick clay landslides, effectively enhancing hazard understanding. Graebling et al. (2024) introduced "VR-EX," a storyline-based VR application for learning geological electrical resistivity tomography experiments at Mont. Terri underground Laboratory (Switzerland), which fostered high engagement, immersion, and knowledge transfer.



As these studies demonstrate, geoscience outreach methods are expanding beyond traditional methods with the integration of innovative technologies and concepts like 3D printing, VFTs, and immersive VR experiences. However, there is still room for further advancement, especially in integrating the lecture component of traditional geoscience education with VR. By combining the interactive nature of VR with the structured learning of lectures, a more dynamic and immersive approach to geoscience education could be achieved, potentially enhancing both engagement, knowledge retention, and ideally, communication to a wider audience.

### 1.2 Objectives

This paper explores the effectiveness of VR-based geoscience lectures in teaching middle school students, addressing the limited focus on this demographic in VR-related geoscience research. Additionally offering an opportunity to introduce the students to geoscience concepts typically inaccessible at their educational level. The study emphasizes geoscience topics related to the 2018 Hokkaido Eastern Iburi Earthquake (HEIE). A fast-paced, portable, VR lecture was developed for this purpose, and its effectiveness is compared to traditional teaching methods. To the author's knowledge, this approach using VR lectures in geoscience education has not been explored before. This study is part of a larger project examining the feasibility of VR in geoscience education. In this paper the focus primarily addresses the motivational and usability aspects of the VR application.

This study narrows on these key objectives: It evaluates the usability of the fast-paced VR lecture format and how it compares to the traditional teaching methods in terms of motivational preference by the participants at this educational level. Furthermore, the study explores the ability of the VR experience to incite interest and curiosity in the geological environment, identifies its strengths and limitations, and outlines potential directions for further development of such teaching approach.

## 2 Methodology

### 2.1 Participants and Setting

The VR outreach program was conducted on the morning of September 19, 2024, at Hokkaido International School (chosen for its diversity of students), running from 9:00 AM to 11:45 AM. The session engaged middle school students (11-14 years old) from grades 6 (G6) through 8 (G8), with a total of 60 participants: 18 from G6, 21 from grade 7 (G7), and 21 from G8. The students came from a variety of international backgrounds, including: East Asia, North America, Oceania, Europe, and Eurasia. This ensures that any bias toward a single perspective in the results is minimized. Hokkaido International School, a private institution, follows a Western-style education system with English as the medium of instruction.

The research workflow consists of three key stages: VR lecture development, survey formulation, and outreach execution. The VR lecture was designed to replicate traditional lectures within a VR environment while leveraging VR-specific advantages, such as visualizing 3D VR models, and animated contents. The survey aimed to evaluate middle school students' motivation and usability experience of the VR lecture, and comparison between traditional



teaching methods. The traditional teaching methods are the lecture and hands-on session. The entire outreach program was strictly limited to 165 minutes by the school and accommodated 60 students.

## 2.2 VR lecture development

This paper defines VR lecture as such only when it meets at least the following key attributes:

- Delivered in a fully immersive 3D VR environment primarily accessed through immersive VR technology
- Designed to teach in a structured manner with defined learning objectives
- Conducted by a lecturer represented as an avatar, featuring voice and expressive body
language
- Participants must have the ability to listen, visualize, navigate using VR-based locomotion, and engage with the lecture and its content within the virtual space
- Incorporates teaching aids that is possible only in VR to enhance understanding
- Capability of being adapted to various instructional settings

The VR lecture is designed in this manner: At the beginning, students will spawn into the starting lobby and is guided to the lecture room after entering a doorway (Fig. 1a). After standing on a designated spot on the floor the lecture will commence. The lecture featured a dummy virtual lecturer delivering a ~5 min oral commentary on the geoscience topics (Fig. 1b). The lecture is fully voiced and showcased virtual teaching aids including images, 360-
degree photospheres, animated models, and 1:1 scale field replica to enhance understanding (Fig. 1c-f).

The 3D models were prepared using photogrammetry in Agisoft Metashape and 3D modelling in Blender. Voice lines were recorded using Microsoft's windows sound recorder application. Unity game engine was used for implementing, animating, and displaying these assets, while
VRChat served as the platform for hosting the lecture during the outreach day. The processing was performed on a desktop PC equipped with an RTX 3080 Ti graphics processing unit (GPU), AMD Ryzen 5 7600x central processing unit (CPU), and 16 GB of random-access memory.



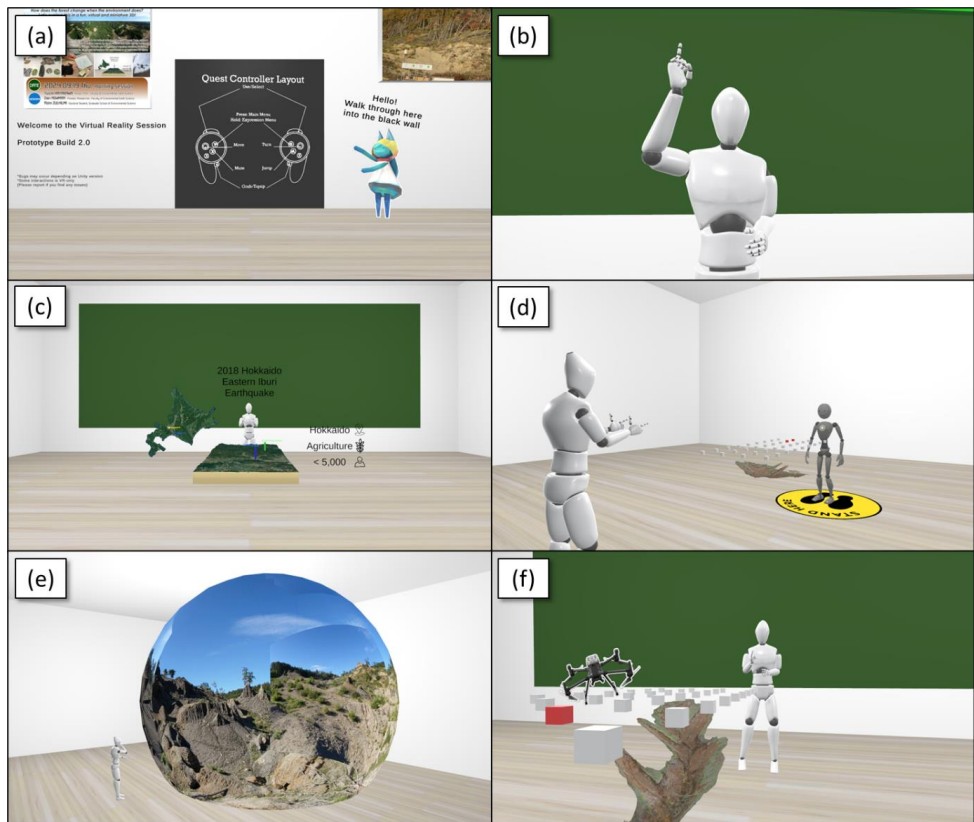

**Figure 1.** World setting for VR lecture (a) the starting lobby room with tutorial board (b) the VR lecturer (c) VR lecturer explaining about the 2018 HEIE (d) participant standing on the trigger zone (e) photosphere showcase of the deep-seated landslide (f) VR lecturer teaching about drones and 3D models

During the prototype phase, which is critical for smooth implementation (Novotny et al. 2019), the VR lecture was tested on graduate students from Hokkaido University's Graduate School of Environmental Science. This process identified technical and content-related issues, allowing the team to refine and finalize the lecture experience. The resulting optimized blueprint was used for the outreach program (Fig. 2).



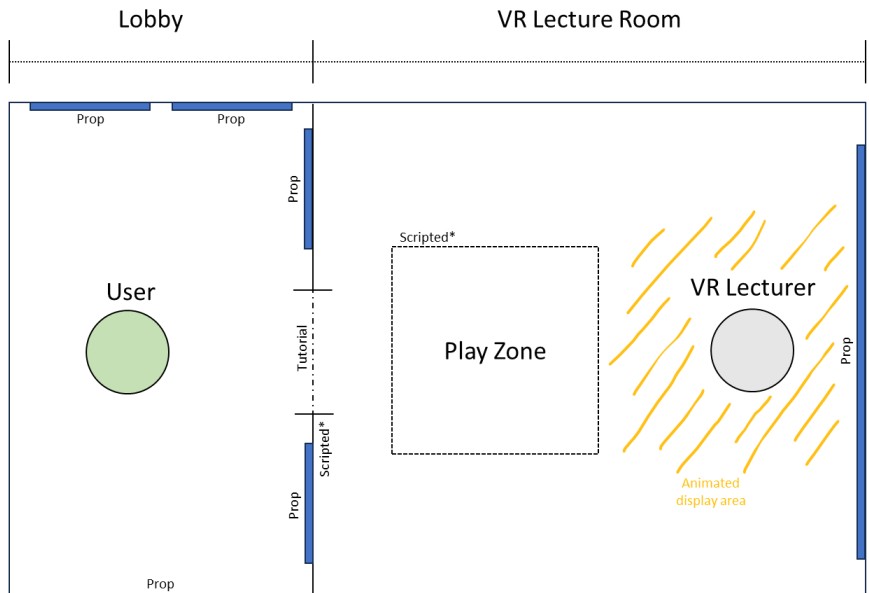

**Figure 2.** VR lecture world blueprint

### 2.3 Survey design

The survey design was inspired by the System Usability Scale (Brooke, 1996), which measures how easy a product is to use, and Intrinsic Motivational Inventory (Ryan & Deci, 2000) for measuring subjective experience of the user in relation to a specific target activity in laboratory experiments. These surveys have been used in previous studies (Carbonell-Carrera et al. 2021; Huang & Liu, 2024; Meulenbroeks et al. 2024).

The author's version of the survey was modified for the prepared VR lecture scenario and adapted to accommodate the time constraints and student amount of the outreach program. After the VR experience, students completed a 10-minute survey consisting of the following sections (refer Appendix A-C):

1.  *VR Usability survey*: This section assessed five categories - Discomfort, Effectiveness, Satisfaction, Immersion, and Accessibility; using a 5-point Likert scale.

2.  *VR/Lecture/Hands-on Motivational survey*: This section evaluated six categories – Interest/Enjoyment, Perceived Competence, Perceived Choice, Effort/Importance, Pressure/Tension, and Value/Usefulness; using a 7-point Likert scale.

3.  *Open Comments*: Students could provide feedback on positives, negatives, and suggestions regarding the session.

To ensure reliable responses, Cronbach's Alpha (Cronbach, 1951; Gliem & Gliem, 2003) was used, with interpretations based on George & Mallery (2002) thresholds ($\geq 0.9$ excellent, $\geq 0.8$ good, $\geq 0.7$ acceptable, $\geq 0.6$ questionable, $\geq 0.5$ poor). The full questionnaires are provided in the Appendix section.



### 2.4 Outreach approach

The outreach program followed a structured schedule: early morning setup, student gathering and briefing, program initiation, and survey data collection (Fig. 3). Equipment deployed included four HMDs (Oculus Quest 2, Meta Quest 2, and two Meta Quest 3 units) supported by an Oculus Link and three third-party link cables. Four PCVR workstations (MSI GS65, HP OMEN, ASUS ROG Zephyrus G14, and HP ZBook Firefly 14 G8) were used, each running Windows OS with integrated CPUs and GPUs.

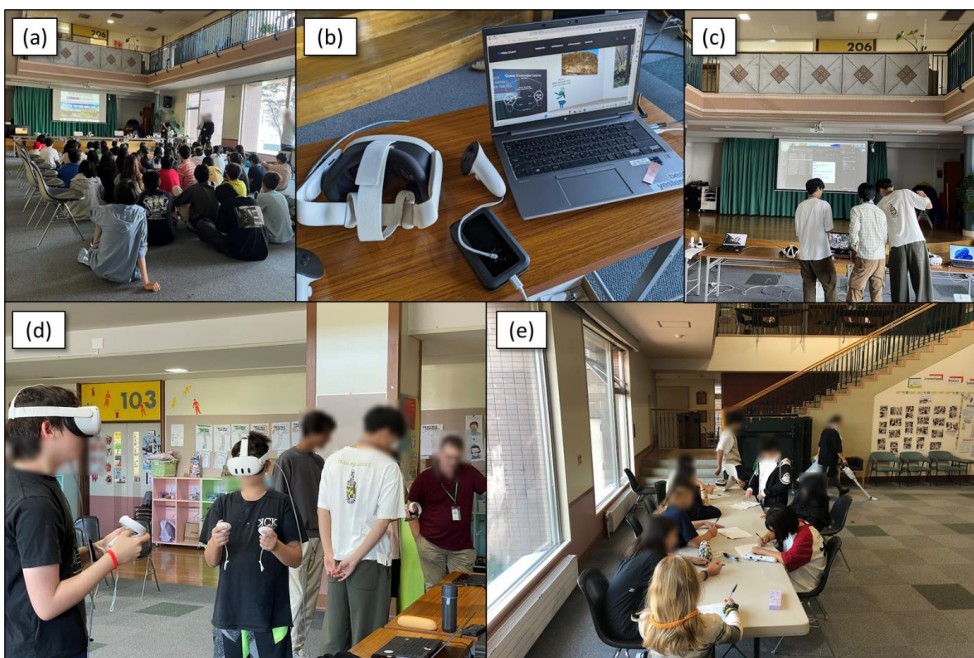

**Figure 3.** Pictures of the VR outreach session. a) students gathering for outreach briefing b) example of a VR workstation c) outreach staff setting up VR stations d) students at the VR stations e) the survey station

The VR sessions were conducted based on the prepared floorplan (Fig. 4). Each session lasted 55 minutes, allowing for three rotations between the three sessions to fit within a total timeframe of 165 minutes. The setup comprised four VR stations, a survey station, and a waiting area. Students were divided into four rows, with each experiencing the VR lecture for approximately five minutes before completing the surveys at the survey station (10 minutes). Those that completed the task moved to the waiting area.



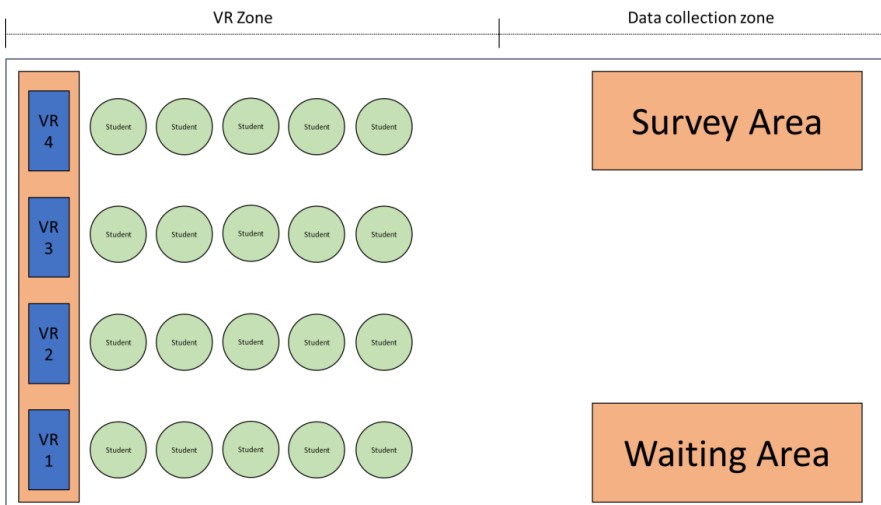

**Figure 4.** The floorplan for the VR outreach program

For other lecture and hands-on sessions, the structure is 40 minutes allocated to activities or lectures and 15 minutes for surveys and transitions. This schedule enabled efficient coverage of all three grades within the allotted time, ensuring a streamlined outreach program.

## 3 Results

### 3.1 VR lecture Usability

The overall Cronbach's alpha value for VR usability was at 0.90, indicating excellent reliability. The VR lecture usability survey results were as follows (Table 1):

- Comfortability: Scored 3.81 (Standard Deviation (SD) = 1.23), reflecting a positive (reverse coded) outcome. This suggests that students disagreed with feeling discomfort during the 5-minute VR lecture.

- Effectiveness: Scored 4.01 (SD = 0.83), indicating a positive outcome. Students agreed that the VR lecture was easy to use and helped them understand the landslide and the authors' research.

- Satisfaction: Scored 4.45 (SD = 0.86), showing an overwhelmingly positive outcome. Students strongly agreed they were satisfied with the VR experience, enjoyed it, and would recommend it to others.

- Immersion: Scored 3.99 (SD = 1.03), resulting in a positive outcome. Students agreed they felt immersed in the experience and wanted to continue beyond the time limit.

- Accessibility: Scored 4.03 (SD = 0.93), demonstrating a positive outcome. Students agreed the VR application provided sufficient support and was easy to use, regardless of prior VR experience.



**Table 1.** *VR Usability Result of Middle School Students*

| Subscale | G6 score | | G7 score | | G8 score | | Total score | |
|---|---|---|---|---|---|---|---|---|
| | M | SD | M | SD | M | SD | M | SD |
| Comfortability | 3.88 | 1.31 | 3.65 | 1.39 | 3.94 | 1.00 | 3.81 | 1.23 |
| Effectiveness | 3.84 | 1.05 | 4.18 | 0.71 | 3.97 | 0.70 | 4.01 | 0.83 |
| Satisfaction | 4.34 | 1.24 | 4.56 | 0.62 | 4.41 | 0.64 | 4.45 | 0.86 |
| Immersion | 4.00 | 1.37 | 4.00 | 0.93 | 3.97 | 0.79 | 3.99 | 1.03 |
| Accessibility | 3.78 | 1.10 | 4.23 | 0.97 | 4.03 | 0.65 | 4.03 | 0.93 |

*Note.* G = Grade; M = Mean; SD = Standard Deviation

### 3.2 Motivational Results

The overall Cronbach's alpha value for the VR lecture motivational survey was at 0.88, indicating good reliability. The survey results are summarized below (Table 2):

- Interest/Enjoyment: Scored 6.18 (SD = 1.27), indicating an overwhelmingly positive outcome. Students strongly agreed that the VR activity was interesting and enjoyable.

- Perceived Competence: Scored 4.97 (SD = 1.41), showing a slightly positive outcome.
Students slightly agreed they understood the session, felt confident in their knowledge, and believed they performed better than their peers.

- Perceived Choice: Scored 5.50 (SD = 1.45), resulting in a positive outcome. Students agreed they participated willingly and felt a sense of freedom during the activity.

- Effort/Importance: Scored 4.44 (SD = 1.61), indicating a slightly positive outcome
(effort item reverse coded). Students slightly agreed they put effort into the session and found the activity meaningful.

- Pressure/Tension: Scored 4.92 (SD = 1.92), showing a slightly positive outcome (both item reverse coded). Students slightly disagreed that the activity caused tension or anxiety.

- Value/Usefulness: Scored 5.23 (SD = 1.36), reflecting a slightly positive outcome. Students slightly agreed the activity was valuable for their future and useful for enhancing geological knowledge, improving their motivation in geoscience.

**Table 2.** *VR Motivational Results of Middle School Students*

| Subscale | G6 Score | | G7 Score | | G8 Score | | Total Score | |
|---|---|---|---|---|---|---|---|---|
| | M | SD | M | SD | M | SD | M | SD |
| Interest/ Enjoyment | 5.71 | 1.93 | 6.42 | 0.72 | 6.33 | 0.79 | 6.18 | 1.27 |
| Perceived Competence | 4.27 | 1.54 | 5.37 | 1.33 | 5.16 | 1.14 | 4.97 | 1.41 |
| Perceived Choice | 5.44 | 1.93 | 5.78 | 1.21 | 5.26 | 1.15 | 5.50 | 1.45 |
| Effort/ Importance | 4.53 | 1.61 | 4.45 | 1.72 | 4.33 | 1.51 | 4.44 | 1.61 |





| Pressure/ Tension | 4.50 | 2.20 | 5.35 | 1.86 | 4.81 | 1.65 | 4.92 | 1.92 |
| Value/ Usefulness | 4.81 | 1.68 | 5.62 | 1.15 | 5.19 | 1.13 | 5.23 | 1.36 |

*Note.* G = Grade; M = Mean; SD = Standard Deviation

The overall Cronbach's alpha value for the lecture session was at 0.86. Indicating good reliability. The survey results are summarized below (Table 3):

- Interest/Enjoyment: Scored 5.39 (SD = 1.36), indicating a positive outcome. Students agreed the lecture was interesting and enjoyable.

- Perceived Competence: Scored 4.67 (SD = 1.44), showing a slightly positive outcome. Students slightly agreed they understood the session, felt confident in their knowledge, and believed they performed better than peers.

- Perceived Choice: Scored 4.86 (SD = 1.15), resulting in a slightly positive outcome. Students slightly agreed they participated willingly and felt some freedom in their choices.

- Effort/Importance: Scored 4.45 (SD = 1.48), indicating a slightly positive outcome (effort item reverse coded). Students slightly agreed they invested effort and found the activity meaningful.

- Pressure/Tension: Scored 5.10 (SD = 1.57), showing a slightly positive outcome (both item reverse coded). Students slightly disagreed that the session caused tension or anxiety.

- Value/Usefulness: Scored 5.18 (SD = 1.51), reflecting a slightly positive outcome. Students slightly agreed the lecture was valuable for their future and useful for enhancing geological knowledge, improving their motivation in geoscience.

**Table 3.** *Lecture Motivational Results of Middle School Students*

| Subscale | G6 Score | | G7 Score | | G8 Score | | Total Score | |
| --- | --- | --- | --- | --- | --- | --- | --- | --- |
| | M | SD | M | SD | M | SD | M | SD |
| Interest/ Enjoyment | 5.46 | 1.21 | 5.52 | 1.20 | 5.19 | 1.62 | 5.39 | 1.36 |
| Perceived Competence | 4.80 | 1.65 | 4.80 | 1.19 | 4.40 | 1.46 | 4.67 | 1.44 |
| Perceived Choice | 5.03 | 1.08 | 5.03 | 1.23 | 4.53 | 1.08 | 4.86 | 1.15 |
| Effort/ Importance | 4.92 | 1.56 | 4.55 | 1.48 | 3.89 | 1.23 | 4.45 | 1.48 |
| Pressure/ Tension | 4.58 | 1.86 | 5.75 | 1.19 | 4.89 | 1.43 | 5.10 | 1.57 |
| Value/ Usefulness | 5.39 | 1.52 | 5.23 | 1.32 | 4.91 | 1.66 | 5.18 | 1.51 |

*Note.* G = Grade; M = Mean; SD = Standard Deviation

The overall Cronbach's alpha value for the hands-on session was at 0.95, demonstrating excellent reliability. On a 7-point Likert scale, the survey results are as follows (Table 4):

- Interest/Enjoyment: Scored 5.40 (SD = 1.27), indicating a positive outcome. Students agreed the hands-on activity was interesting and enjoyable.





- Perceived Competence: Scored 4.95 (SD = 1.25), showing a slightly positive outcome. Students slightly agreed they understood the session, felt confident in their knowledge, and believed they performed better than peers.

- Perceived Choice: Scored 4.91 (SD = 1.33), resulting in a slightly positive outcome. Students slightly agreed they participated willingly and felt some freedom in their choices during the hands-on.

- Effort/Importance: Scored 4.51 (SD = 1.61), indicating a slightly positive outcome (effort item reverse coded). Students slightly agreed they invested effort and found the activity meaningful.

- Pressure/Tension: Scored 4.91 (SD = 1.77), showing a slightly positive outcome (both item reverse coded). Students slightly disagreed that the session caused tension or anxiety.

- Value/Usefulness: Scored 5.21 (SD = 1.11), reflecting a slightly positive outcome. Students slightly agreed the hands-on activity was valuable for their future and useful for enhancing geological knowledge, improving their motivation in geoscience.

**Table 4.** *Hands-on Motivational Results of Middle School Students*

| Subscale | G6 Score | | G7 Score | | G8 Score | | Total Score | |
|---|---|---|---|---|---|---|---|---|
| | M | SD | M | SD | M | SD | M | SD |
| Interest/ Enjoyment | 5.39 | 1.50 | 5.73 | 1.07 | 5.07 | 1.18 | 5.40 | 1.27 |
| Perceived Competence | 5.18 | 1.45 | 4.95 | 1.21 | 4.73 | 1.07 | 4.95 | 1.25 |
| Perceived Choice | 4.97 | 1.53 | 5.48 | 1.30 | 4.18 | 0.67 | 4.91 | 1.33 |
| Effort/ Importance | 4.58 | 1.80 | 4.58 | 1.65 | 4.37 | 1.35 | 4.51 | 1.61 |
| Pressure/ Tension | 4.18 | 2.04 | 5.83 | 1.22 | 4.50 | 1.57 | 4.91 | 1.77 |
| Value/ Usefulness | 5.39 | 1.23 | 5.35 | 1.04 | 4.79 | 0.92 | 5.21 | 1.11 |

*Note.* G = Grade; M = Mean; SD = Standard Deviation

### 3.3 Open comments

The open comments provided valuable insights into the positive and negative traits of each session as perceived by the participants. Not to be mistaken for concrete frequency data pertaining these traits but only what was noticeable and mentioned by the participants. Students' feedback ranged from single words to more detailed responses, which required the author to code them individually according to a keyword-based guideline (Table 5). The frequency of responses for each code reflects the traits that were most mentioned, highlighting the strongest impressions of the sessions. While even a single mention indicates a positive or negative trait, the most frequently cited elements point to the most prominent experiences of the participants.

**Table 5.** *Coding procedure based on keywords*

| Code | Example keywords |
|---|---|
| Accessibility | Difficulty, freedom, mobility, clarity, familiarity |





| Content | New knowledge, examples, materials, chapters, 3D materials, information, explanation, summary page, figures |
|---|---|
| Discomfort | Sickness, eye pain, cumbersome |
| Engagement | Style, fun, interesting, cool, enjoyable, satisfying, entertaining, interactive, positivity, delivery |
| Immersion | Presence, realism, visuals |
| Inaccessibility | Difficulty, lack possession, confusion, difficulty, equipability |
| Insufficient content | Lack content, simplicity, lack pictures, lack activity, lack explanation |
| Interaction | Hands-on, classroom interaction |
| Management | Distractions, lack instructions |
| Novelty | New experience, unique |
| Poor Engagement | Zoning out, boring, awkward, lack of interactions, lack participation, sitting, lack freedom, repetitive, passive |
| Technical Issues | Audio, bugs, glitches, lagging |
| Technology | VR device, drone |
| Time | Time limit, waiting time, long speech, long lecture |
| Value | Experience, usefulness, exposure |

In terms of positive aspects, the VR session received the most positive feedback for its engagement, followed by the content, immersion, technology, accessibility, novelty, and interaction (Table 6). The lecture session was most appreciated for its content, with engagement, accessibility, value, and interaction also receiving positive mentions. Similarly, the hands-on session was praised for its content, engagement, and interaction, followed by technology, accessibility, and value. The primary takeaway from the VR session was its ability to engage students, while the lecture and hands-on sessions were most valued for their content.

**Table 6.** *Open comments on positive aspects of the sessions*

| Code | VR (from 47 response) | Lecture (from 45 response) | Hands-on (from 37 response) |
|---|---|---|---|
| Accessibility | 4 | 10 | 2 |
| Content | 11 | 28 | 22 |
| Engagement | 32 | 17 | 12 |
| Immersion | 8 | 0 | 0 |
| Interaction | 2 | 3 | 10 |
| Novelty | 3 | 0 | 0 |
| Technology | 7 | 0 | 8 |
| Value | 0 | 4 | 1 |

On the other hand, the negative aspects identified in the feedback revealed areas for improvement. For the VR session, the most frequently mentioned negative aspects included technical issues, discomfort, inaccessibility/management, insufficient content, and time constraints (Table 7). For the lecture session, the main concerns were inaccessibility, poor engagement, insufficient content, and time limitations. The hands-on session received similar negative feedback, with inaccessibility, poor engagement, and insufficient content being the



most common complaints. The key negative issue for the VR session was the presence of technical problems, while the lecture and hands-on sessions were predominantly affected by accessibility challenges.

**Table 7.** *Open comments on negative aspects of the sessions*

| Item | VR (from 30 response) | Lecture (from 26 response) | Hands-on (from 13 response) |
|---|---|---|---|
| Inaccessibility | 5 | 13 | 8 |
| Insufficient content | 3 | 4 | 1 |
| Discomfort | 7 | 0 | 0 |
| Management | 5 | 0 | 0 |
| Poor Engagement | 0 | 10 | 4 |
| Technical Issues | 13 | 0 | 0 |
| Time | 2 | 2 | 0 |

## 4 Discussion

### 4.1 VR usability impacts

Overall feedback on VR-related discomfort was positive (Table 1), with most middle school students reporting no significant issues during the 5-minute lecture. However, seven students experienced discomfort, including eye strain, headset heaviness, and motion sickness (Table 7). Motion sickness, or cybersickness, is a known challenge in VR adoption (Chang et al. 2020; Chattha et al. 2020; Keshavarz et al. 2011). The positive experience likely resulted from the short lecture duration, minimal movement, simple design, and optimized performance, all of which could have reduced discomfort triggers. Previous studies indicate that longer sessions, complex tasks, and unrealistic locomotion increase motion sickness risk due to sensory conflict (Dużmańska et al. 2018; Reason & Brand, 1975; Saredakis et al. 2020). By keeping the experience brief and straightforward, these issues were effectively minimized.

The effectiveness of the VR experience received positive feedback (Table 1), with students finding it easy to use, navigate, and effective for learning geoscience topics. The lecture was designed to be simple and time-efficient, requiring minimal interaction; students only needed to stand, look, and listen. Open comments confirmed that students gained new knowledge, with 3D models being particularly helpful for understanding and visualizing discussed concepts (Table 6). Research on penetrative thinking (Bagher et al. 2020) and spatial abilities (Gittinger & Wiesche, 2024) supports that VR can benefit low spatial ability learners. Therefore, strategically designed VR lectures with 3D models can enhance geoscience teaching and improve learning effectiveness.

The satisfaction category received overwhelmingly positive feedback, with students across all grades enjoying the VR experience (Table 1). Open comments emphasized the excitement of learning geoscience in an engaging way, enhanced by a variety of 3D geological examples (Table 6). Similar positive responses to VR in educational settings have been reported across age groups, from high school to university (Graebling et al. 2024; Visneskie et al. 2020;





Yamauchi et al. (2022). These findings compliments those literature, suggesting that VR is well-received by students, even at the middle school level.

The immersion category also received positive feedback, with students feeling absorbed and expressing a desire for longer VR sessions (Table 1). The VR lecture's immersive design, featuring a scripted virtual lecturer, detailed 3D content, and animations, likely enhanced this effect. Students highlighted the visuals, sense of "being there," and freedom of movement as
key contributors to their sense of immersion (Table 6). Similar findings have been observed in geological VR research, including simulations, role-playing, and visualization (Alene et al. 2024; Graebling et al. 2024; Klippel et al. 2019). These results support that VR effectively immerses students, fostering a sense of presence with the geoscience content.

The accessibility category received positive feedback, with students finding the VR lecture
easy to use despite having no prior experience. The simple tutorial (hypothesized to pose some difficulties for students), featuring a single panel with basic instructions (Fig. 1a), was generally sufficient, though G6 students were more neutral, possibly needing additional guidance or time to consult it (supplementary data). Some negative feedback pointed to the tutorial's simplicity and lack of detail (Table 7). This aligns with research highlighting the need for a robust
familiarization phase (Harknett et al. 2022; Papadopoulou et al. 2022; Wright et al. 2023). While the current tutorial was adequate for most of the students, improvements for younger audiences, like G6, are recommended. Future studies should explore ways to enhance the tutorial or familiarization process for younger audiences.

Overall, the VR usability of this VR lecture experience was positive with a strong level of
satisfaction. With rooms for improvements for discomfort, effectiveness immersion, and accessibility to obtain a stronger opinion.

### 4.2 VR motivational impacts

The VR session outperformed the lecture and hands-on sessions in Interest/Enjoyment, receiving overwhelmingly positive feedback (Table 2, 3, and 4). Its immersive 3D content and
interactive experience led to strong engagement, with no reports of poor engagement. This aligns with similar studies in geoscience education (Graebling et al. 2024; Visneskie et al. 2020). However, critiques of the VR session primarily focused on its limited duration. Lectures and hands-on sessions were praised for content delivery, with lectures benefiting from teacher interaction and hands-on sessions excelling in interactivity. Traditional methods, however,
faced issues like poor engagement and excessive session length (Table 7), which could induce boredom (Mann & Robinson, 2009). While VR's immersion is effective for engagement, the novelty may diminish with repeated use, warranting further studies.

All sessions showed slightly positive outcomes in Perceived Competence, with no significant preference among students (Table 2, 3, and 4). Likely, VR's 3D models and immersive content
supported comprehension and spatial reasoning (Bagher et al. 2020; Gittinger & Wiesche, 2024). Students reported that VR aided mental visualization of geological concepts, though the absence of subtitles, lack of interactivity, and fast pacing affected younger students (G6). Lectures achieved positive outcomes for their content and clarity, but some students desired



more in-depth clarifications. Hands-on sessions benefited from interactive 3D models and
drones but occasionally struggled with complex content. To improve confidence and
comprehension, VR could incorporate aiding elements (e.g. subtitles), interactive features, and
slower pacing, while adjusting content based on grade level.

Perceived Choice was rated positively for all sessions, with VR slightly outperforming others
(Table 2, 3, and 4). Students appreciated VR's novelty and the freedom to explore models from
different angles. Individual presence in virtual worlds can foster a sense of freedom and agency,
allowing users to explore and interact with objects in ways impossible in the physical world
(Chirico et al. 2018). Lectures offered interaction with instructors, and hands-on sessions
excelled in exploration and drone demonstrations. VR's lower score in this category stemmed
from limited content, non-interactive models, and short session durations, compounded by
having only four VR stations. VR's high cost could limit classroom implementation if resources
are scarce. Nonetheless, VR's immersive environment provided a unique sense of autonomy,
motivating engagement and fostering independence. Longer, more interactive sessions are
recommended, possibly by increasing VR workstation availability.

All sessions received slightly positive outcomes for Effort/Importance, indicating slight
importance without being too effort demanding (Table 2, 3, and 4). VR was praised for
simplifying content with 3D materials but may have been seen more as a fun tool than a serious
learning medium, reducing its perceived importance (Table 6). Lectures were appreciated for
clear explanations and future potential relevance, while hands-on sessions excelled in
interactivity but likely required more effort to understand the small-scale geology models.
Emphasizing educational value in VR and incorporating more detailed interactive elements
could enhance perceived effort and relevance.

Pressure/Tension outcomes were slightly positive across all sessions, reflecting minimal stress
or anxiety (Table 2, 3, and 4). VR's immersive environment and low complexity may contribute
to a relaxed experience, this aligns with research on virtual presence (Pavic et al. 2023).
However, issues like peer interference and the 5-minute time limit occasionally caused
frustration. Lectures and hands-on sessions benefited from engaging materials and supportive
instructors, though content density with complexity could have contribute to increased
perceived tension. Addressing timing, peer distractions, and content difficulty may reduce
pressure and improve the perception of the VR lecture.

Finally, all sessions scored slightly positive for Value/Usefulness, with students recognizing
benefits in geoscience knowledge and motivation but not finding them highly impactful (Table
2, 3, and 4). VR was praised for enhancing visualization of landslides through 3D models, but
the short duration and limited real-world applications diminished its impact. Lectures were
valued for content and clarity, while hands-on sessions excelled in interactivity. The
entertainment-oriented perception of VR, stemming from its origins in the entertainment
industry (Havenith et al. 2019; Hornsey & Hibbard, 2024), may have influenced their
perception outcomes. Future studies should connect geoscience concepts to practical
applications and extend VR session durations to increase perceived value and usefulness.



### 4.3 Advantages and disadvantages of the VR lecture

From the study it revealed that there are several advantages and disadvantages of delivering VR lectures in a school setting most evident from the open comment entries by the participants (Table 6 & Table 7) and the results of the survey (Table 1, 2, 3, & 4).

The project demonstrated VR's advantages for geoscience education. The fast-paced VR lecture was enough to engage middle school students showcasing its ability to attract and motivate effectively. Its short duration minimized VR-related discomfort. Its immersive nature offered a safe exploration of hazardous sites, enhanced understanding through 3D models, and bridged abstract and practical concepts not easily achievable through traditional means. Additionally, the novelty of VR technology in the geoscience curriculum excited and appealed to the students, while the freedom to explore the virtual environment, including free-roaming capabilities and interaction with the VR system, further enriched their learning experience.

Despite its strengths, the VR lecture presented several challenges. Although majority of students did not experience discomfort, there were still some that did (e.g., motion sickness, eye strain, HMD weight), and technical issues like individuals changing audio on shared headsets, glitches, and lag were common. VR experience can also vary depending on the developers' expertise, time, and budget, posing a potential challenge for standardized implementation. VR is also not yet widely adopted in current educational settings and requires time for users to learn the system for efficient usage. High costs limit headset availability, necessitating shared use, which leads to shorter sessions and extended wait times. Addressing these issues requires thoughtful planning, resources, and financial support especially considering implementation for geoscience education.

### 4.4 Limitations of the approach and future studies recommendations

It is advised that readers understand the limitations of this work as it will be helpful when using it as a reference.

Researcher bias could influence the interpretation of qualitative open comments, despite adherence to coding guidelines (Table 5), as student responses varied from vague to detailed. Session delivery styles differed among chairs, potentially affecting student experiences. The survey was adapted for a young audience and time constraints, limiting its comprehensiveness; future studies with fewer constraints could address this. While the sample was international, it lacked true global representation. These difference in a countries' learning culture may influence how individuals learn (Joy & Kolb, 2009). Not all students completed the questionnaire, especially open comment sections, due to factors like unwillingness to provide feedback or logistical issues. Although the VR session had the highest response rate, no session achieved full participation. Lastly, the study's fast-paced design, while efficient for collecting data, compromised survey depth and the VR experience quality, which could have been enhanced with additional time.

Future research should expand the sample to include more countries, capturing cultural differences and offering a broader global perspective. Investigating the perceptions of students with extensive VR experience could provide insights into how familiarity influences outcomes,



potentially through repeated exposure experiments to mitigate the novelty effect over time. Another promising avenue is testing VR in multiplayer settings, where students collaborate to learn geological concepts. Additionally, live, teacher-led VR lectures could be compared with scripted formats to explore how educators adapt to VR and how students benefit versus traditional methods. Hybrid approaches combining traditional and VR-based learning also

warrant further exploration.

## 5 Conclusions

This study developed a 5-minute, fast-paced, automated VR lecture to evaluate its usability and motivational impacts on international middle school students learning geological topics. To the best of the author's knowledge, it is the first known attempt to assess VR-based geoscience

lectures for this demographic, it also showcases an efficient method for gathering data within a limited timeframe. Results demonstrated that the VR lecture was usable, with students reporting high satisfaction levels. Motivational impacts were positive, excelling in fostering interest and enjoyment, and perceived choice. Overall, students showed a clear motivational preference for VR over traditional teaching methods.

While the VR geoscience lecture was successful, there remains room for improvement, particularly in enhancing its usability and motivational outcomes. Future research should address the limitations identified in this study to achieve consistently strong verdict across all survey categories. Additionally, insights from challenges in traditional teaching methods can also be used to guide the further refinement of VR implementation in middle school education.

This study highlights the potential of VR as a potential platform for engaging diverse audiences and disseminating geoscientific knowledge.

### Data availability

The expanded versions of the data used in this study are available as supplementary materials.

### Authors contribution

All authors contributed to the study conception, design, material preparation, and data collection. The first draft of the manuscript was written by AZ and all authors commented on previous versions of the manuscript. All authors read and approved the final manuscript.

### Competing Interests

The authors declare no competing interests.

### Ethical Statement

This study was conducted with the head of school consent, adhering to ethical guidelines. Any personal data collected was anonymized, and participation was voluntary, prioritizing students' well-being and educational benefit.




**Acknowledgements**

The authors would like to acknowledge the support of the Ministry of Education, Culture, Sports, Science and Technology (MEXT) of the Japanese government for providing the scholarship that sponsored this study.

The authors extend their gratitude to Tim Schlosser, Head of School, and middle school teachers Nick Fazio and Tricia Calhoon, along with the dedicated staff members of the school, for their invaluable support and coordination in successfully initiating this project.

We also gratefully acknowledge the support provided by students Masafumi Inomata, Tomoki Onodera, and Shunsaku Matsumura for their valuable assistance.

**Financial Support**

This work is partially supported by Japan Society for the Promotion of Science (JSPS) KAKENHI Grant Numbers 23K20541, JP21H00627, 23KF0180, JSPS Bilateral Program Number JPJSBP120233201, and 2023 Hokkaido University COI-NEXT fund.





**Appendix A: Usability survey**

Student ID: ___________

# VR Usability (VR Activity)

For each entry below, **circle the response** that best characterized how you feel about the statement.

Discomfort

|  | Strongly Disagree | Disagree | Neutral | Agree | Strongly Agree |
|---|---|---|---|---|---|
| I felt some discomfort (e.g. sickness, eye strain etc.) | 1 | 2 | 3 | 4 | 5 |

Effectiveness (ease of learning and using)

|  | Strongly Disagree | Disagree | Neutral | Agree | Strongly Agree |
|---|---|---|---|---|---|
| It was easy to use and navigate in VR | 1 | 2 | 3 | 4 | 5 |
| VR helped me understand the landslides and research work conducted in Atsuma better | 1 | 2 | 3 | 4 | 5 |

Satisfaction

|  | Strongly Disagree | Disagree | Neutral | Agree | Strongly Agree |
|---|---|---|---|---|---|
| I was very satisfied with the VR experience | 1 | 2 | 3 | 4 | 5 |
| I would recommend the VR activity to other students | 1 | 2 | 3 | 4 | 5 |
| I liked the VR experience | 1 | 2 | 3 | 4 | 5 |





Engagement

| | Strongly Disagree | Disagree | Neutral | Agree | Strongly Agree |
|---|---|---|---|---|---|
| I wanted to continue the VR activity beyond the time limit | 1 | 2 | 3 | 4 | 5 |
| During the VR activity, it felt like I was really there | 1 | 2 | 3 | 4 | 5 |

Accessibility

| | Strongly Disagree | Disagree | Neutral | Agree | Strongly Agree |
|---|---|---|---|---|---|
| The VR application provided sufficient help and support | 1 | 2 | 3 | 4 | 5 |
| I easily understood how to use the features of VR regardless of prior VR experience | 1 | 2 | 3 | 4 | 5 |





**Appendix B: Motivational survey**

Student ID: ___________

# Motivational Survey (__ Activity)

For each entry below, **circle the response** that best characterized how you feel about the statement.

Interest/ Enjoyment

|  | Strongly Disagree | Disagree | Slightly Disagree | Neutral | Slightly Agree | Agree | Strongly Agree |
|---|---|---|---|---|---|---|---|
| The __ activity was fun & entertaining | 1 | 2 | 3 | 4 | 5 | 6 | 7 |
| The __ activity was enjoyable & satisfying | 1 | 2 | 3 | 4 | 5 | 6 | 7 |
| The __ activity was interesting & appealing | 1 | 2 | 3 | 4 | 5 | 6 | 7 |


Perceived competence

|  | Strongly Disagree | Disagree | Slightly Disagree | Neutral | Slightly Agree | Agree | Strongly Agree |
|---|---|---|---|---|---|---|---|
| I understood majority of the __ session | 1 | 2 | 3 | 4 | 5 | 6 | 7 |
| I felt I performed better than my peers | 1 | 2 | 3 | 4 | 5 | 6 | 7 |
| I felt more confident and capable about my newly acquired knowledge after the __ activity | 1 | 2 | 3 | 4 | 5 | 6 | 7 |



Perceived choice

|  | Strongly Disagree | Disagree | Slightly Disagree | Neutral | Slightly Agree | Agree | Strongly Agree |
|---|---|---|---|---|---|---|---|
| I was involved with the __ activity because I wanted to | 1 | 2 | 3 | 4 | 5 | 6 | 7 |
| I felt I had some freedom of choice during the __ activity | 1 | 2 | 3 | 4 | 5 | 6 | 7 |

Effort/Importance

|  | Strongly Disagree | Disagree | Slightly Disagree | Neutral | Slightly Agree | Agree | Strongly Agree |
|---|---|---|---|---|---|---|---|
| I had to put a lot of effort during the __ activity | 1 | 2 | 3 | 4 | 5 | 6 | 7 |
| It is important to me to do well during the __ activity | 1 | 2 | 3 | 4 | 5 | 6 | 7 |

Pressure/Tension

|  | Strongly Disagree | Disagree | Slightly Disagree | Neutral | Slightly Agree | Agree | Strongly Agree |
|---|---|---|---|---|---|---|---|
| The __ activity felt tense | 1 | 2 | 3 | 4 | 5 | 6 | 7 |
| I felt anxious during the __ activity | 1 | 2 | 3 | 4 | 5 | 6 | 7 |






Value/Usefulness

| | Strongly Disagree | Disagree | Slightly Disagree | Neutral | Slightly Agree | Agree | Strongly Agree |
|---|---|---|---|---|---|---|---|
| The __ activity will value me in the future | 1 | 2 | 3 | 4 | 5 | 6 | 7 |
| The __ session is useful for my geological knowledge | 1 | 2 | 3 | 4 | 5 | 6 | 7 |
| This __ experience improved my motivation in Geology | 1 | 2 | 3 | 4 | 5 | 6 | 7 |








**Appendix C: Open comments**


Student ID: ____________

# Open entry (__ Activity)

- In your opinion, what are the positive parts of the __ experience?



- In your opinion, what are the negative parts of the __ experience?



- Please provide some suggestion to improve the __ activity and experience




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
