# Peer review of "Usability and motivational impact of a fast-paced immersive virtual reality lecture on international middle school students in geoscience education"

_EGUsphere, 2025_

## Author Response (AR1)

**Response to Reviewer Comments**

**Reviewer #1**: This is an interesting study that offers an excellent commentary on the practicalities, benefits, and limitations of using VR for geoscience communication activities. The methodology of the study allows for a robust evaluation of the effectiveness of the activity in comparison to traditional (e.g. lecture-based) educational delivery. I particularly value the detail of the study setup provided through Section 2 to allow others to broadly replicate the activity. Overall, I recommend publication of this manuscript with some minor changes:

1. The introduction / rationale for the study should include further references to literature that demonstrates a VR-lecture approach – e.g. Jong et al. (2020), Hagge (2024), and Harknett et al. (2022) (Full references below). The study rightly outlines that there is a clear gap in the literature for the evaluation of VR-based pedagogies, but there has been some work in this field that goes above and beyond that presented in Lines 73 – 80.

   - Thank you very much for suggesting these articles. We have included these articles in the literature review section and integrated them into the rationale paragraph.
   - These are very great additions to provide more information and context for the readers.
     - Page 2, Line 75-77
     - Page 3, Line 85-86, 89-92

2. The development of the activity (lines 142 – 147) is good and this level of specificity is very helpful – did you choose to custom design this activity because there were no ready-to-go apps that suited this purpose? It might be helpful for a reader looking to replicate your activity to understand the decision-making behind putting extensive effort in to this design.

   - Thank you for the comment and inquiry. Yes, we designed it ourselves, partly because no existing apps met our specific needs and partly due to the constraints of limited time and a large pool of participants. As a result, a custom experience had to be developed.
   - This workflow may be useful for readers planning a time-limited study while gathering a substantial sample size.
   - We have added a brief introductory paragraph about this in Section 2.2.
     - Page 4, Line 135-139

3. The presentation of results through Section 3.2 is good, but a table that compares the total scores between the VR, lecture, and hands-on activity would be beneficial to allow an easier comparison between the activities.

- As suggested, we added the table for the convenience of the readers at the end of Section 3.2.
  - o Page 11, Line 305-307
  - o Page 12, Line 309

4. Following on from (2), it'd be interesting to hear some further reflection (beyond what is suggested in lines 378 – 379) on the lack of a significant difference in perceived competence. This opens a broader question that many VR-based pedagogy studies have seldom addressed – is the time and cost effort of creating VR activities worth it if attainment and understanding of taught concepts is largely the same? Some reflection from the authors on this point (perhaps in Section 4.3) would help to draw out the purpose and value of bringing VR into education spaces.

- This is a good point! To address this comment, we added an additional paragraph about our stance on this in section 4.3 of the discussion.
- In summary, we agree that it can be a significant barrier in adoption. But still, it offers additional benefits that traditional methods cannot. It provides various innovative ways to, and for the first time, effectively communicate the true 3D nature and scale of geoscience concepts to students. Perhaps, such advantages garner potential for long-term benefits such as improved retention and increased motivation.
- Still, we raised some very important concerns in Section 4.4, that acts as a pathway to guide further studies hoping that our fellow community can build upon and contribute as a collective.
  - o Page 17, Line 482-495

5. The conclusion notes that students had a motivational preference for VR teaching (line 478), but then subsequently that the activity could be made better to improve motivation (line 481) – please just check these sentences for this contradiction!

- In this section, we intended to convey that the middle school students preferred VR when compared to traditional methods. Regarding the second sentence, our aim was to highlight the potential for further improving the VR lecture architecture to yield stronger conclusions.

- Pardon the writing, your comment is actually a good highlight, and we appreciate it, as it could potentially lead to confusion for future readers. To address this, we have reworded the sentences for clarity and improved readability.
  - Page 18, Line 529-532

**Reviewer 2#**: This is a timely article given the increased interest in and use of VR in geosciences teaching. The authors guide us through the current state of the research in this field, and outline the challenges and opportunities posed by adopting VR in 'classroom-based' teaching. The approach they have adopted in novel, with most users of VR taking more of a field guide rather than virtual lecturer approach. The methods chosen to assess/evaluate the effectiveness of the approach are robust. There is a lot of critical reflection by the authors in terms of their data and the meaning, as well as an awareness of their positionality in the process. I would recommend this be accepted with minor revisions.

1. The methodology section needed a little more detail on the traditional lecture and activities (section 2.4, line 200). It wasn't clear what those activities (or hands-on session) were or how they might engaged the learners. It was also not clear whether activities and/or a lecture was part of the approach in section 2.4, but it was clearly mentioned as an approach in line 121. Some clarification for the reader to make it clear would be helpful here.

   - Thank you for the suggestion. To address this comment and provide greater clarity for readers, we have added details about the traditional activity sessions and how they engaged learners at the end of Section 2.4.
     - Page 8, Line 215-218

2. Whilst the focus of the article is on the motivational and usability of VR in geosciences teaching, it might be helpful to include/reflect on the potential learning gain of utilising VR instead of traditional approaches to teaching. I appreciate this cannot be measured based on the research, but this is something that VR has the potential to improve for some learners. This could be mentioned in the literature review, and there are a number of recent (since 2020) systematic reviews/meta-analyses that have addressed learning outcomes from VR in middle schools/their equivalents.

   - Thank you for the suggestion. This is a good point! In general, VR does enhance students' knowledge gain. We have added additional citations from reviews and meta-analyses in the Introduction, before the Literature Review, as we feel this placement is more fitting to provide a smoother transition and improve the overall flow toward the next section.
     - Page 2, Line 47-50

3. It was great to read about the critiques/limitations that the researcher encountered, but I wonder if there is more scope for including some reflections on the time/effort aspects

of this approach to teaching and learning. Producing content for VR is more than just writing a lecture, it is all the associated technical work and planning required. This has been raised in publications around developing virtual fieldwork resources/digital approaches to teaching. Given the time/effort component of the developmental work by the authors, there is scope for a few sentences on this aspect as it is useful to acknowledge that 'unseen' aspect.

- It is true that producing content for VR lectures is significantly different from preparing traditional lectures. Based on our experience, it requires preparation and skills beyond simply having knowledge of geoscience subject matter.
- To address this suggestion, we added a paragraph in Section 4.3 as an advice for future readers to be aware of such challenges.
  - Page 17, Line 475-481

4. With VR approaches to teaching, equity and inclusion are issues that are not fully acknowledged/addressed here. When looking forwards to the application of such approaches it is important to acknowledge who might be excluded and who has access to this technology? Also, to consider the impacts on staff in terms of development of the resources (see point 3). There is scope for acknowledgement of some of the issues (challenges) around VR in the classroom.

- Thank you for this suggestion. You are correct that, while VR expands access to geoscience education for some groups, such as individuals with mobility impairments, it still does not fully accommodate everyone for example, those who are visually impaired, and may remain more accessible to more affluent individuals.
- To address this point, we have acknowledged this in Section 4.3.
  - Page 17 & 18, Line 467-471

5. Just some very minor typos/formatting: Line 41 suggest replacing 'the masses' with all. Line 100 could the objectives be distilled into obj. (i) and obj. (ii)? Lines 113-115 assuming I have not misunderstood that a 'private institution' is an independent (fee-paying) school, then could a sentence acknowledging that there could be a 'class' bias in the sample population be added? Line 136 should that be ARE and not is guided? Line 200 should that be 'For THE other lecture...'?

- Thank you for identifying the typos and providing further suggestions. We very much appreciate it and fully agree with all the recommendations.
  - Page 1, Line 41

- Page 3, Line 108-115
- Page 4, Line 151
- Page 8, Line 215-218
- Page 17, Line 501-502

**Summary**

We would like to express our gratitude to the referees, Dr. Liam Taylor and Dr. Lynda Yorke, for their valuable feedback, which greatly helped to enhance the manuscript. We addressed the comments and updated the content of the manuscript accordingly.

The manuscript revision involved additional reflections in the Discussion section, acknowledgment of further limitations, and clarifications to aid reader understanding. As a result, we incorporated the suggested recommendations. We also appreciate the reviewers for pointing out typos and potential clarity issues, which we have addressed. We have further addressed all inquiries in a point-by-point manner.

**Manuscript change log**

*Introduction*

- Added new citations to the introduction
  - Page 2, Line 47-50
- Added new citations to the literature review
  - Page 3, Line 75-77, 85-86, 89-92
- Improved word choice/style
  - Page 1, Line 41

*Methodology*

- Updated grammar
  - Page 4, Line 151
- Added an additional paragraph to detail the context for the methodology
  - Page 4, Line 135-139
- Added additional information regarding traditional teaching sessions
  - Page 8, Line 215-218

*Results*

- Added Table 5 for the convenience of readers regarding total motivational results
  - Page 11, Line 305-308
  - Page 12, Line 308-309

*Discussion*

- Corrected in-text citation format
    - Page 14, Line 367
- Associated in-text citation to exact table in supplementary materials
    - Page 14, Line 381
- Added a new supportive citation
    - Page 15, Line 425-427
- Added two additional paragraphs to Section 4.3
    - Page 17, Line 475-495
- Added some statement to acknowledge challenges regarding inclusivity to Section 4.3
    - Page 16, Line 467-471
    - Page 17, Line 472
- Added a sentence to acknowledge class bias of private institutions
    - Page 17, Line 501-502
- Added transition words to improve reading flow
    - Page 17, Line 501-502, 503, 505, 507

*Conclusion*

- Improved the readability of the conclusion
    - Page 17, Line 529-532

*References*

- Added eight additional references
- Adjusted the placement of one reference by alphabetical order

*General*

- Improved the format and formality of writing expressions in some paragraphs throughout the manuscript